# Effects of Tribulus (*Tribulus terrestris* L.) Supplementation on Erectile Dysfunction and Testosterone Levels in Men—A Systematic Review of Clinical Trials

**DOI:** 10.3390/nu17071275

**Published:** 2025-04-06

**Authors:** José de Oliveira Vilar Neto, Wilson Max Almeida Monteiro de Moraes, Daniel Vieira Pinto, Carlos Alberto da Silva, Juan de Sá Roriz Caminha, Júlio César Chaves Nunes Filho, Caio Eduardo Gonçalves Reis, Jonato Prestes, Heitor O. Santos, Elizabeth De Francesco Daher

**Affiliations:** 1Graduate Program in Medical Sciences, Federal University of Ceará, Fortaleza 60355-636, Brazil; jvilarr@gmail.com (J.d.O.V.N.); danielvieirapinto@gmail.com (D.V.P.); juanroriz2@gmail.com (J.d.S.R.C.); juliocesaref@yahoo.com.br (J.C.C.N.F.); ef.daher@yahoo.com.br (E.D.F.D.); 2Physical Education and Sports Institute, Federal University of Ceará, Fortaleza 60355-636, Brazil; carlosas@ymail.com; 3Graduation Program on Physical Education, Catholic University of Brasília, Brasília 71966-700, Brazil; jonatop@gmail.com; 4Department of Nutrition, School of Health Sciences, Universidade de Brasília, Brasília 70910-900, Brazil; caioedureis@gmail.com; 5School of Medicine, Federal University of Uberlandia, Uberlândia 38408-100, Brazil; heitoroliveirasantos@gmail.com

**Keywords:** *Tribulus terrestris*, hypogonadism, testosterone, sexual dysfunction, saponins

## Abstract

**Background**: *Tribulus terrestris* L. Zygophyllaceae (TT) is a plant that has been claimed to increase testosterone levels and improve sexual function, particularly erectile dysfunction, with potential benefits for male sexual health. **Purpose**: This systematic review aimed to evaluate the effectiveness of TT supplementation in improving sexual function and serum testosterone levels in men. **Methods**: We conducted a systematic review adhering to the Preferred Reporting Items for Systematic Reviews and Meta-Analyses (PRISMA) guidelines. After searching the literature (*n* = 162), 52 studies were selected for full-text reading, and 10 studies were eligible for this review, comprising 9 clinical trials and 1 quasi-experimental study (a study without a control). The Jadad score revealed low methodological quality for 50% of the studies. **Results**: The studies involved 15 to 172 participants (total = 483) aged between 16 and 70 years with different health conditions: healthy men (*n* = 5), oligozoospermia (*n* = 1), erectile dysfunction (*n* = 1), erectile dysfunction associated with hypogonadism (*n* = 2), and unexplained infertility (*n* = 1). TT supplementation at doses of 400 to 750 mg/d for 1 to 3 months improved erectile dysfunction in 3 of the 5 studies that assessed this parameter. Eight out of ten studies did not report significant changes in androgen profile following TT supplementation, but the subjects in the neutral studies did not have low androgen levels at baseline. Therefore, only 2 studies showed significant intra-group increase in total testosterone levels, which had low clinical magnitude (60–70 ng/dL) and involved subjects with hypogonadism. **Conclusions**: TT supplementation has a low level of evidence regarding its effectiveness in improving erectile function in men with erectile dysfunction, and no robust evidence was found for increasing testosterone levels.

## 1. Introduction

Male hypogonadism is an androgen insufficiency disorder marked by low total testosterone levels (<300 ng/dL), which is the main biochemical parameter for diagnosis [1]. The burden of male hypogonadism has grown worldwide, affecting of 10 to 40% of men, depending on diagnostic criteria [2]. This phenomenon is not only associated with energy loss, fatigue, decreased physical capability, and impaired body composition, but also with erectile dysfunction (ED), a clinical parameter that markedly affects male health [3]. Apart from traditional hormone replacement therapy in men [4], many non-pharmacological strategies have emerged in an attempt to mitigate androgen deficiency, with no discernable effects documented to draw solid clinical recommendations [5,6].

The plant *Tribulus terrestris* L. Zygophyllaceae (TT) has been used for centuries, mainly in India and China, as an herbal medicine and as a sexual stimulant to treat ED and augment testosterone levels. As TT is native to warm, temperate and tropical regions, it is not only found in India and China, but also in the Mediterranean, Bulgaria, Mexico, and some regions of the US. Approximately 20 species of TT are documented worldwide [6,7].

Some studies indicate that TT supplementation modulates androgen profile through an increase in serum testosterone levels and the testosterone/estradiol ratio [8,9]. Apparently, steroidal saponins are the main TT phytochemical compound responsible for the rise in the levels of testosterone and related hormones, such as luteinizing hormone, dehydroepiandrosterone, and dehydroepiandrosterone sulphate. However, the mechanism by which TT elevates testosterone levels is not fully understood [4]. Saponins from TT also act directly as a neurosteroid, increasing dehydroepiandrosterone levels, which may exert an antagonistic effect on gamma aminobutyric acid (GABA) and thereby enhance sexual function regardless of testosterone levels [7]. Moreover, an upregulation of nitric oxide (NO) synthesis induced by TT supplementation is a supposed mechanism to enhance erectile function due to its vasodilator effect [10].

Although a couple of studies report efficacy of TT supplementation on the androgen profile and sexual function of men with infertility, ED, and older men [8,9], other evidence indicates that TT supplementation is not effective for these purposes [8,11]. In addition, TT extracts are commonly taken as supplements by athletes of different sports, with no evidence for improvement in hormonal profile and sexual function [12,13].

Given the relevance of the topic and the recent studies showing promising results regarding the potential benefits of TT supplementation for ED [14], the present study aimed to summarize the literature about the effects of TT supplementation on testosterone levels and erectile dysfunction. We hope that the results of this study will be relevant, as they can contribute to scientific progress in the field and provide knowledge of the effects of TT supplementation on male hormonal profile and sexual function according to academic literature.

## 2. Methods

This study was registered in the international prospective register of systematic reviews database (PROSPERO) with the protocol number CRD42020202273. The study design employed PRISMA guidelines (Preferred Reporting Items for Systematic Reviews and Meta-Analysis) [15].

### 2.1. Search Strategy

A literature search in 13 databases was performed without language or date restrictions (Table 1) between April and June of 2024 with the following terms: “*Tribulus terrestris*” OR “Tribulus” AND “Testosterone” OR “Androgens” OR “Male Infertility” OR “Male Hypogonadism” OR “Erectile dysfunction”. Further information about the search strategy (search terms, filters, and conditions) and data management is presented in Table 1.

### 2.2. Study Eligibility Criteria

The study eligibility criteria involved clinical trials and case reports that encompassed men submitted to TT supplementation alone and that assessed serum androgen levels or sexual function. Men of any age and heath condition were considered. The PICOS eligibility strategy was applied [16]. Further details about study eligibility and the selection process are shown in Table 2 (PICOS framework) and in the PRISMA flowchart diagram (Figure 1). Unpublished documents, review studies, conference abstracts, theses, book chapters, comments, and response articles were excluded. Studies examining women or animals and those with any other drug or substance that could interfere with androgen profile or sexual function were excluded.

### 2.3. Study Selection

The literature search and eligibility process were performed by two independent authors (JOV and DVP). The screening process of the databases was based on the title and abstract to assess eligibility. Duplicate articles were identified and removed. After that, the screened papers had the full text reviewed, and those suitable for the review were selected. At this stage, the articles were excluded with given reasons. Disagreements were solved through consultation with a third author (HOS), an expert in the field.

### 2.4. Risk of Bias

The methodological quality of each study was evaluated using the Jadad scale, a 5-point scale which assignes scores for reported randomization, blinding, and withdrawals [17]. After that, the selected studies were considered high quality with a score above 3 out of 5.

### 2.5. Data Extraction

Data were extracted by author JOV and were double-checked by author DVP to avoid errors. A third author (HOS) solved disagreements. The following information was extracted from the selected studies: author, publication year, study origin and design, subject characteristics, TT supplementation protocol, androgen profile, sexual function and any other outcomes, collateral effect, and Jadad score.

## 3. Results

The search results are presented in the PRISMA flowchart (Figure 1). The literature search process, based on the PICOS strategy, identified a total of 162 results from 13 scientific databases (Table 1 and Figure 1). Out of 162 results, 48 were identified as duplicates. After the removal of the duplicates, 114 results remained for the detailed analysis. After searching the abstracts in these 114 results, 24 of these results were unpublished studies, 10 were animal models, 21 were in women, 4 were theses, 1 was a congress abstract, and 1 was a response letter. Consequently, these 61 results were excluded.

Thus, 52 studies remained for a complete reading. After reading and fully analyzing these 52 studies and considering the inclusion and exclusion criteria (Table 2), 43 studies were excluded for the following reasons: (i) study design was not a clinical trial in 22 studies; (ii) other drugs or substances were administered along with TT in 14 studies; (iii) 6 studies did not evaluate serum androgen profile or sexual function, and (iv) 1 study was in the Russian language. Finally, 10 studies were qualified and selected for the presented systematic review (Figure 1).

Table 3 summarizes the studies included in the systematic review. Studies were from Brazil (1 study), Australia (1 study), India (1 study), Egypt (3 studies), Bulgaria (2 studies), China (1 study), and Spain (1 study). The subjects were aged from 16 to 70 years with the following health conditions: 5 studies with healthy men, 1 study with oligozoospermia, 1 study with ED, 2 studies with ED associated with hypogonadism, and 1 study in men with unexplained infertility. The sample size ranged from 15 to 172 participants. No study reported side effects.

Regarding the TT supplementation protocol, the dosage regimens varied from 400 mg to 12 g/day for 4 weeks to 3 months. According to the Jaded score, three studies presented high methodological quality [12,22,23]: 4 points to Santos et al. 2014 [22] and Rogerson et al. [12] and 5 points to Kamenov et al. [23]. Therefore, the majority of the studies (*n* = 5) presented poor quality (≤3 points).

Concerning the main outcomes, all studies investigated the effects of TT supplementation on serum androgen levels. Of these, two studies [9,21] supported an increase in testosterone levels; they included men suffering from hypogonadism. As to ED, TT supplementation was investigated in five studies and elicited benefits in three of them. Interestingly, all 3 studies [9,19,21] reporting benefits included patients with ED, while the null result study [11] included patients with oligozoospermia, not with ED.

## 4. Discussion

The present systematic review evaluated the effects of TT supplementation on testosterone levels and ED. Overall, TT supplementation improved erectile function in men with ED. Regarding the dosage regimen for ED, TT supplementation varied from 400 to 750 mg/d for 1 to 3 months in patients with mild-to-moderate ED [9,12,21,23]. Furthermore, TT supplementation did not increase testosterone levels, while these studies did not focus on men with hypogonadism [8,11,12,13,18,23,24,25]. Few studies reported a significant increase in testosterone levels with low clinical magnitude (~60–70 ng/dL) in men with hypogonadism [9,21].

The highest dose of TT supplementation tested was 12 g/d in patients with oligozoospermia [11]. In general, both TT and placebo groups had improved parameters of ED, penile weakness and loss of rigidity, and premature ejaculation, while testosterone levels did not change significantly. Furthermore, this study reported benefits of TT supplementation on semen analysis and safety at an uncommonly high dosage. To the best of our knowledge, a feasible plateau effect cannot be established to date, mainly considering that the study with the highest dose of TT has several biases and received 2 points for the Jadad score. Thus, we advance that further well-controlled RCTs are essential in the attempt to provide a reliable dose–response curve with applicability to the clinical scenario.

Despite the proposed effects of TT supplementation as a testosterone booster through different mechanisms, 80% of the studies analyzed in this systematic review did not report significant changes in the androgen profile following TT supplementation (400–750 mg/d for 2–3 months) [8,10,12,13,18,19,22]. However, only two studies exclusively enrolled subjects with low testosterone levels (<350 ng/mL) and observed effects of TT supplementation as a testosterone booster [9,21]. In both studies, subjects received 3 capsules daily (750 mg) of TT (Trib Gold, origin Bulgaria, 250 mg of TT and a minimum of 45% of saponins per capsule) over the course of 3 months. More specifically, through a randomized, single-blind, placebo-controlled trial, 70 patients with late-onset hypogonadism reported a ~58 ng/mL (27%) increase in mean total testosterone levels (from ~215 to ~273 ng/dL, *p* < 0.001) in the group receiving TT [8]. Similarly, Roaiah et al. [21] reported a significant 71 ng/dL (33%) increase in mean total testosterone levels (from ~213 to ~284 ng/dL, *p* < 0.01) with TT supplementation in middle-aged and older men with partial androgen deficiency. Indeed, both studies presented considerable methodological bias, rating 2 points [9] and 0 score [21] in Jadad analysis.

In summary, the aforementioned studies [9,21] support a potential ~60–70 ng/dL increase in total testosterone levels following TT supplementation in men with low testosterone levels at baseline within a context of poor methodological quality. Regardless of this concern, such a testosterone increase is equivalent to the potential of vitamin D supplementation and some herbals supplements, such as long Jack (*Eurycoma longifolia Jack*, Simaroubaceae) and fenugreek (*Trigonella foenum-graceum* L., Fabaceae) [5,25]. Conversely, this potential is inferior to other herbal medicines such as mucuna (*Mucuna pruriens* (L.) DC., Fabaceae) and ashwagandha (*Withania somnifera* (L.) Dunal, Solanaceae), as well as therapeutic doses of zinc and arginine in men with low testosterone and related urological disorders [5,25,26].

Despite inconclusive evidence for TT supplementation as a testosterone booster, TT presented advantageous effects for patients suffering from ED in 3 of the 5 studies [9,19,21]. The study by Sellandi et al. [11] reported that the TT treatment (12 g/d of TT supplementation) for 60 days was not able to improve ED as compared with placebo. Despite this, the authors showed an increase in motility and sperm morphology. These results are in concordance with a previous systematic review [27], highlighting TT supplementation as effective in improving sperm parameters and potentially reducing infertility.

GamalEL et al. [9] and Roaiah et al. [8], who investigated ED and unexplained infertility, respectively, observed that 1500 mg/d of TT supplementation for 3 months significantly increased the mean score of the international index of erectile function (IIEF-5), while no effects were reported for placebo. Furthermore, Kamenov et al. [19] revealed that 500 mg/d of TT supplementation over the course of 3 months improved IIEF-5 scores as well as intercourse satisfaction, sexual desire, orgasmic function, overall satisfaction, and global efficacy questionnaire (GEQ) responses. Differences between TT supplementation and placebo became statistically significant (*p* = 0.0119) even after only 4 weeks of treatment and continued to increase progressively toward the end of the treatment period (*p* < 0.0001). These data corroborate the findings by Santos et al. [22], who showed that 400 mg/d of TT supplementation over the course of 4 weeks improved IIEF-5 scores. Collectively, these results suggest that a minimum dosage regimen of 400 mg/d of TT for 4 weeks improves ED.

Among the studies demonstrating improvements in sexual function parameters following TT supplementation, one of them [19] reported no effect regardless of the increase in sex hormone levels. Although sexual dysfunction and androgen deficiency often coexist, improvements in sexual function and clinical signs of hypogonadism do not necessarily co-occur [28].

As previously mentioned, saponins are the biologically active compounds of TT responsible for an upregulation of nitric oxide synthesis [10], thereby improving the nitric oxide synthase pathway, endothelium-dependent relaxation, consequent penile corpus cavernosum relaxation, and finally, erectile function [19]. It is worth mentioning that saponin concentration differed from 40 to 60% of TT supplements between the included studies, with the exception of the study by Santos et al. [22], which reported no details of saponin content. Thus, not only do saponin concentrations differ by geographic region, but different dosages resulting from pharmacy TT compounding manipulations must also be considered in the clinical scenario [29].

Apart from the main outcomes of this systematic review, it is worth mentioning that neutral effects of TT supplementation on sports performance and body composition (Table 3) were observed. At best, TT is a low-cost supplement and well-tolerated at the documented dosage regimens, as no clinical side effects or impairment in lipid or liver panel tests were registered in the studies (Table 3), revealing the potential safety of TT supplementation [30].

This systematic review suggests that further studies are needed to consider TT supplementation as a coadjutant strategy to treat ED. Therefore, TT supplementation is not a substitute for pharmacological and surgical interventions used as first-line treatments for severe ED conditions. Nevertheless, caution is mandatory to not extrapolate TT supplementation as a testosterone booster, as we did not find solid evidence for this scenario.

As for limitations, a high risk of bias was detected for 50% of the studies included in this systematic review. Therefore, further well-controlled clinical trials are imperative to portray reliable conclusions about TT supplementation in the field of male health and hence to afford qualified trials for future meta-analysis studies. Given a sharp selection bias and unreliable data description in the original trials, performing a meta-analysis was avoided, and this study focused on a systematic review associated with a critical appraisal of the studies, while at the same time discussing potential benefits for the clinical scenario. There was an express concern about the studies that assessed testosterone levels since null results were obtained from studies that evaluated patients with normal testosterone levels [8,11,12,13,18,19,20,24], while positive results came from studies performed with patients with low testosterone levels (studies with crucial limitations). These studies [9,21] reported an increase in testosterone levels (60–70 ng/dL) with a low clinical magnitude effect; therefore, this result can likely be replicated in real-world scenarios and future studies.

## 5. Conclusions

TT supplementation (400 to 750 mg/d for 1 to 3 months) can improve erectile function in men suffering from mild-to-moderate ED. To date, there is no solid evidence that TT supplementation is a testosterone booster. Importantly, studies that reported null effects of TT supplementation on testosterone levels did not focus on patients with low testosterone. Thus, further well-controlled, randomized clinical trials are warranted to establish an effective TT supplementation protocol (dosage and duration) to improve erectile function. In addition, it could be explored whether TT supplementation could be an adjuvant strategy to raise testosterone and related androgen levels in men with hypogonadism. 

## Figures and Tables

**Figure 1 nutrients-17-01275-f001:**
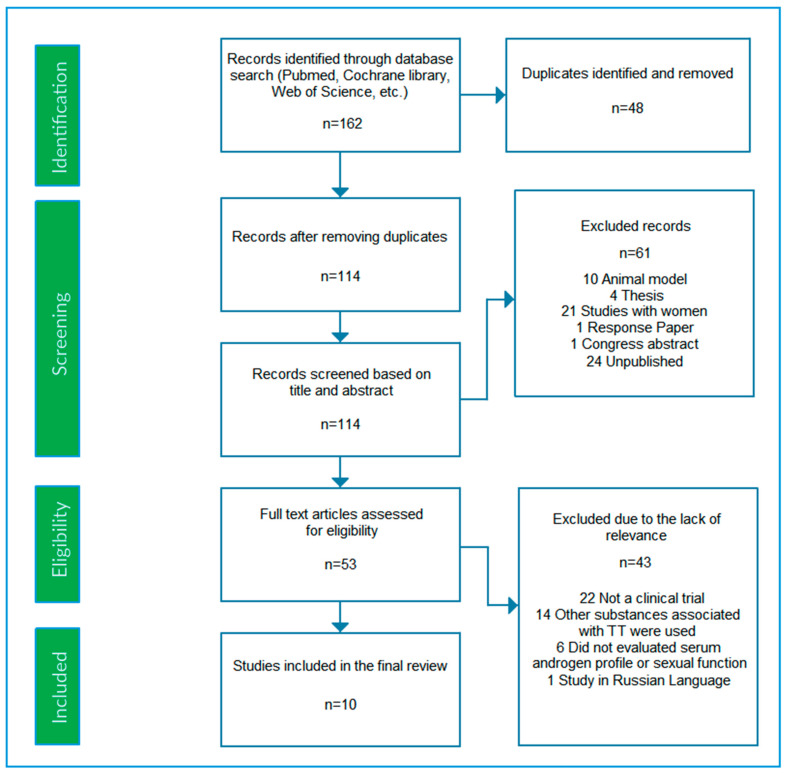
PRISMA flow diagram of the systematic review search results.

**Table 1 nutrients-17-01275-t001:** Databases and websites searched (including search terms and conditions) from inception between April and June 2024.

	Database or Website	Search Terms, Filters and Conditions	Found
**1**	PubMed	(*Tribulus terrestris*) [Title] Filters: Case Reports; Clinical Study; Clinical Trial; Randomized Controlled Trial	20
**2**	ClinicalTrials.gov	Other Terms: *Tribulus terrestris* Filters: Completed Studies; Studies with Results; Studies with Male Participants	0
**3**	PMC	Included in PubMed Search	-
**4**	MedLine	Included in PubMed Search	-
**5**	OpenGrey Repository	*Tribulus terrestris*	4
**6**	American Chemical Society Publications	Title: Hypogonadism	20
**7**	ScienceDirect	Terms: *Tribulus terrestris* testosterone Title, abstract, keywords: *Tribulus terrestris* testosterone Filters: Research articles, Case Reports	11
**8**	Web Of Science	(Tribulus [Title]) AND Terrestris [Title] AND Testosterone [Title]) Filters: Articles, English.	2
**9**	Lilacs	Included in PubMed Search	-
**10**	EU Clinical Trials	*Tribulus terrestris* Filters: Trials Completed; Trials with results; Male.	1
**11**	SCOPUS	TITLE (tribulus AND terrestris AND testosterone) AND ( LIMIT-TO (PUBSTAGE, “final”)) AND (LIMIT-TO (DOCTYPE, “ar”))	3
**12**	Cochrane Library	Title, Abstract or Keyword: “*Tribulus terrestris*” Filters: Trials	91
**13**	Google Scholar	allintitle: *Tribulus terrestris* testosterone (Excluded patents and citations)	9

**Table 2 nutrients-17-01275-t002:** Study eligibility criteria—PICOS framework.

PARAMETER	CRITERIA
POPULATION	Men
INTERVENTION	*Tribulus terrestris* (only)
COMPARISON	Control or Placebo group
OUTCOMES	Serum androgen profile and/or sexual function
STUDY DESIGN	Clinical trial and case report.

**Table 3 nutrients-17-01275-t003:** Eligible studies: study, subjects, TT protocols, and outcome description.

References	Origin	Study Design	Subject	TT Protocol	Improved Serum Androgen Profile?	Improved Sexual Function?	Other Outcome	Collateral Effect?	Jadad Score
**[18]**	Bulgaria	Randomized, placebo-controlled clinical trial.	21 healthy men aged 20–36 years old.	Daily dose was 20 and 10 mg/kg body weight per day of TT (60% of SS), separated into three daily intakes for 4 weeks.	No. Serum total testosterone, androstenedione and LH concentrations were not significantly altered after TT intake.	Not investigated.	Not investigated	Not reported	1
**[9]**	Egypt	Randomized, placebo-controlled clinical trial.	70 men aged 40–70 years old, with ED and partial androgen deficiency (total testosterone below 350 ng/dL).	3 capsules daily (total 750 mg) of TT at least 45% of SS per capsule) three times daily after meals for 12 weeks.	Yes. Serum total testosterone was significantly higher after TT intake.	Yes. General scores of IIEF-5 showed significantly increased after TT intake.	Serum PSA-t and AST levels were higher after treatment. No changes in serum ALT or IPSS.	Not reported	2
**[19]**	Bulgaria	Multicenter, phase-IV, prospective, randomized, double-blind, placebo-controlled clinical trial.	172 male subjects aged 18–65 years with mild or moderate ED for at least 6 months and/or secondary HSDD.	6 tablets daily (total 1500 mg) of TT (Tribestan, Sopharma AD, 250 mg of TT not less than 45% of SS per tablet) orally, three times daily after meals for 12 weeks.	No. The results for all parameters (total testosterone, free testosterone, DHEA-S, and SHBG) did not differ between groups.	Yes. TT intake improved erectile function, intercourse satisfaction, orgasmic function and sexual desire (assessed by IIEF-5).	No statistical differences in serum total cholesterol, triglycerides, LDL, or HDL levels. In addition, no difference was found in blood pressure parameters.	Not reported	5
**[20]**	China	Randomized, placebo-controlled clinical trial.	15 healthy male boxers, aged mean of 16 years old. Minimum of two years of boxing training experience.	2 capsules daily (total 1250 mg) of TT (Pronova Biocare Co, 625 mg and above 40% SS per capsule) orally, one intake every morning for 6 weeks. All subjects were submitted to same training protocol, with 3 week high-intensity training and three week high-volume training.	No. TT did not significantly alter plasma levels of testosterone, dihydrotestosterone, or IGF-1. However, IGFBP3 was decreased by TT.	Not investigated.	No statistical differences in TMM or TFM. TT intake ameliorated muscle damage and promoted aerobic performance.	Not reported	3
**[8]**	Egypt	Randomized placebo controlled clinical trial.	30 male patients, aged 30–50 years, with unexplained infertility for more than a year with no obvious cause.	3 capsules daily (total 750 mg) of TT orally for 12 weeks.	No. TT intake did not change serum total testosterone, free testosterone, or LH values. No statistically significant difference was found between groups.	Not investigated.	TT intake did not change sperm concentration, motility, or abnormal forms parameters.	Not reported	1
**[21]**	Egypt	Quasi-experimental non-controlled.	30 male patients, aged 40–70 years, with ED, low libido, and total testosterone below 12 nmol/L (345.8 ng/dL).	3 capsules daily (total 750 mg) of TT orally, three times daily, for 12 weeks.	Yes. Serum levels of total testosterone and free testosterone were increased significantly after TT intake. LH levels showed no difference after the treatment.	Yes. General scores of IIEF-5 showed significantly increased after TT intake.	Not investigated	Not reported	0
**[12]**	Australia	Randomized, placebo-controlled, double-blind clinical trial.	22 healthy male rugby athletes aged 19.8 ± 2.9 years. Minimum of 12 months of resistance training experience.	1 capsule daily (total 450 mg) of TT (60% SS) for 5 weeks. All subjects were submitted to same 5-week strength and conditioning program.	No. TT intake did not change urinary testosterone/epitestosterone ratio. No statistical differences were found between groups.	Not investigated.	TT did not improve maximal strength and body composition. Both groups had higher parameters in 2 RM strength evaluation. In addition, no statistical differences were noted in BMI or FFM.	Not reported	4
**[22]**	Brazil	Randomized, placebo-controlled, double-blind clinical trial.	30 men aged mean of >40 years old with ED.	2 capsules daily of TT (400 mg of the dry extract each capsule) for one month.	No. Serum total testosterone was not altered by TT intake.	No. None difference between groups in General scores of IIEF-5.	Not investigated	Not reported	4
**[11]**	India	Randomized, placebo -controlled, double-blind clinical trial.	63 men aged 21–50 years with oligozoospermia (sperm count below 20 million/mL).	TT (herb) in granule at 2 daily intake of 6 g before food (12 g/day) with warm water for 60 days.	No. TT intake did not change serum testosterone, FSH, or LH. There was no statistically significant difference in intragroup or intergroups analyses.	No. Both groups had an increase in parameters like ED, penile weakness and loss of rigidity, and premature ejaculation. However, just the treated group had improved post-act exhaustion parameters.	TT-treated group showed an increase in motility of sperm and less abnormal forms.	Not reported	2
**[13]**	Spain	Randomized, single-blind, placebo-controlled trial	30 healthy CrossFit^®^-trained males aged 30–50 years. Minimum of 20 months of CrossFit experience		No. TT intake did not change serum testosterone, cortisol, and testosterone/cortisol ratio. There is no statistical difference in intragroup or intergroups analyses.	Not investigated.	TT did not alter body composition, performance, mood, or perceived exertion	Not reported	3

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
