# Peer review of "Effects of Tribulus (*Tribulus terrestris* L.) Supplementation on Erectile Dysfunction and Testosterone Levels in Men—A Systematic Review of Clinical Trials"

_nutrients, 2025, doi:10.3390/nu17071275_

Round 1

Reviewer 1 Report

Comments and Suggestions for Authors

L21: avoid 1st person language

L42: you suggest that it has grown, can you provide any incidence or prevalence data? Or changes over time?

L44 and 62: looks like some extra spaces

L50: due to its proposal to treat (seems grammatically awkward)

.....stimulant.  It has been proposed as a supplement to treat ED and.....

L74: avoid 1st person language

L94: why include case reports?

L131: 1 was a congress abstract, and 1 was a response....

L143: I will be curious to read why to include healthy men?  Did these healthy men have lower TT levels?

Table 3: can it be stretched

L173: At least this study (seems too conversational in its writing style)

L194: aforementioned (not above-mentioned)

L240: avoid 1st person

L247: I hadn't seen previously where there was a discussion about including other pharmacologic aids?  This seems out of the blue and probably should not be included then

L258: Don't start sentence with Because

L259 and 261: avoid 1st person

Author Response

We appreciate your suggestions. The text of the manuscript has been revised and the changes are in green.

We remain at your disposal for any questions you may have.

  • L21: avoid 1st person language

Response: Done. Introduction lines 2 and 3

L42: you suggest that it has grown; can you provide any incidence or prevalence data? Or changes over time?

Response: Done. Introduction. Lines 3 and 4.

L44 and 62: looks like some extra spaces

Response: Done.

L50: due to its proposal to treat (seems grammatically awkward)

.....stimulant.  It has been proposed as a supplement to treat ED and.....

Response: Adjusted page 2, lines 5-7.

L74: avoid 1st person language

Response: unidentified

L94: why include case reports?

Response: Deleted case reports (Results line 10).

L131: 1 was a congress abstract, and 1 was a response....

Response: Modified (Results line 7)

L143: I will be curious to read why to include healthy men?  Did these healthy men have lower TT levels?

Response: not necessarily. As the idea was to compile all clinical trials to have a better view of the effects of Tribulus including patients with various characteristics, healthy men were included.

Table 3: can it be stretched

Response: Done.

L173: At least this study (seems too conversational in its writing style).

Response: Replaced by furthermore, this study… L173.

L194: aforementioned (not above-mentioned).

Response: Done. L 194.

L240: avoid 1st person

Response: Modified L240

L258: Don't start sentence with Because

Response: modified by since L258

L247: I hadn't seen previously where there was a discussion about including other pharmacologic aids?  This seems out of the blue and probably should not be included then

Response: Removed from the text, as well as references.

L259 and 261: avoid 1st person

Response: Done. L259 and 261

Reviewer 2 Report

Comments and Suggestions for Authors

This is an interesting study which adds to an area where there is little research despite high interest from the general public.

The review is well conducted and well-reported. The only change required is an updating of the search. The results reported come from 2022, now 3 years ago. The search needs to be updated to establish whether there is any additional information.

Comments on the Quality of English Language

There are some areas where the language can be improved. For example the use of the phrase "a couple" eg line 167 should be changed to 2. The phrase "a couple" has a colloquial tone and is not used interchangeably with 2 in everyday English.

Author Response

We welcome comments and suggestions will be considered.

We reviewed the English in the text and believe it has improved.

This is an interesting study which adds to an area where there is little research despite high interest from the general public.

The review is well conducted and well-reported. The only change required is an updating of the search. The results reported come from 2022, now 3 years ago. The search needs to be updated to establish whether there is any additional information.

Response: Thank you for commentaries. The text said “2022”, but the investigation went on until 2024. Text modified  in topic Search strategy and title of table 1 in yellow.

There are some areas where the language can be improved. For example the use of the phrase "a couple" eg line 167 should be changed to 2. The phrase "a couple" has a colloquial tone and is not used interchangeably with 2 in everyday English.

Response: text modified “Few studies” line 167 in yellow

The points appointed for revisers are in green (rev 1) and yellow (rev 2).

Reviewer 3 Report

Comments and Suggestions for Authors

THis is systematic review of the effect of Tribulus Terrestris L. Zygophyllaceae (TT)  on erectile dysfunction and testosterone level in men.Results are not convincing in regard of any effect on observed parameters.

In Table 1, only 3 studies reported improvement in ED; while authors claim in text of Discussion that 4 studies demonstrated improvement in ED, please explain. lines 155-156

As author stated, studies on testosterone level impact of TT are not convincing. Also, studies on effects on ED are also doubtfull. Please adjust your Discussion and conclusions accordingly.

Author Response

THis is systematic review of the effect of Tribulus Terrestris L. Zygophyllaceae (TT)  on erectile dysfunction and testosterone level in men.Results are not convincing in regard of any effect on observed parameters.

Response: Thank you for considerations. We reviewed the data in table 1 and the description was adjusted.

Modified in text “in three of them. Interestingly, all 3 studies [9, 21, 23]” in gray.

In Table 1, only 3 studies reported improvement in ED; while authors claim in text of Discussion that 4 studies demonstrated improvement in ED, please explain. lines 155-156

Response: Modified “…in five studies and elicited benefits in three of them” line 155

and “all 3 studies [9, 21, 23] ” line 156

As author stated, studies on testosterone level impact of TT are not convincing. Also, studies on effects on ED are also doubtfull. Please adjust your Discussion and conclusions accordingly.

Response: Thank you for commentaries. We adjusted the title of the manuscript for Effects of Tribulus (Tribulus terrestris L.) supplementation on erectile dysfunction and testosterone levels in men – a systematic review of clinical trials.

The conclusion in abstract: Conclusion: TT supplementation has a low level of evidence regarding its effectiveness in improving erectile function in men with erectile dysfunction, and no robust evidence was found for increasing testosterone levels.

And in Discussion: This systematic review suggest that further studies are needed to consider TT supplementation as coadjutant strategy to treat ED.
